# Genebanks at Risk: Hazard Assessment and Risk Management of National and International Genebanks

**DOI:** 10.3390/plants12152874

**Published:** 2023-08-04

**Authors:** Theresa Herbold, Johannes M. M. Engels

**Affiliations:** 1Faculty of Agricultural Sciences, University of Hohenheim, 70599 Stuttgart, Germany; 2Alliance of Bioversity International and CIAT, 00153 Rome, Italy; j.engels@cgiar.org

**Keywords:** genebanks, plant genetic resources, hazard assessment, natural hazards, political risks, risk management, risk prevention, risk mitigation, risk transfer, insurance

## Abstract

Genebanks are crucial for safeguarding global crop diversity but are themselves exposed to several risks. However, a scientific basis for identifying, assessing, and managing risks is still lacking. Addressing these research gaps, this study provides risk analysis for three key risk groups: natural hazards, political risks, and financial risks, carried out on a sample of 80 important national and international genebanks, comprising at least 4.78 million accessions or roughly 65% of the reported total of ex situ conserved accessions worldwide. The assessment tool of Munich Re “Natural Hazards Edition” allowed a location-specific comparison of the natural hazard exposure. Results showed that genebanks in the Asia-Pacific region are most exposed to natural hazards, while institutions in African and some Asian countries are rather vulnerable to political risks. Financing is a major problem for national genebanks in developing countries, whereas the Global Crop Diversity Trust achieved considerable financial security for international genebanks. Large differences in the risk exposure of genebanks exist, making a location- and institution-specific risk assessment indispensable. Moreover, there is significant room for improvement with respect to quality and risk management at genebanks. Transferring risks of genebanks to third parties is underdeveloped and should be used more widely.

## 1. Introduction

In the context of protecting agrobiodiversity, the conservation of plant genetic resources (PGR) has become an important pillar whereby ex situ conservation realized off-site in genebanks is the principal approach [1,2]. In about 1750 genebanks and collections, nearly 7.4 million plant accessions are maintained worldwide [3].

Genebanks are essential for conserving and making available PGR not only for current progress in plant breeding but also as a treasure for use by future generations. Despite the growing worldwide awareness of this potential, it is often overlooked that genebanks themselves are subject to several challenges and risks that might jeopardize their physical integrity. Past incidents and events resulting in the partial loss of important crop collections or even entire genebanks have shown this drastically. A recent example is the ICARDA genebank, which was originally located in Aleppo, Syria. As a consequence of the Syrian civil war and severe combat operations in Aleppo starting in 2012, the genebank had to be relocated in 2016 to Lebanon and Morocco. Part of the germplasm collection could be restored, with safety duplicates preserved at international genebanks and at the Svalbard Global Seed Vault (SGSV) in Spitsbergen, Norway [4,5]. In 2011, the national genebank of Thailand was flooded, which caused the loss of some of the 20,000 unique rice accessions maintained there [5]. The national genebank of the Philippines at Los Baños was damaged by flooding due to a typhoon in 2006 and hit again by a fire in 2012 [5,6]. Ukraine’s seed bank in Kharkiv was at high risk as the city has been a staging ground of military operations since the start of the Russian invasion of Ukraine in February 2022 [7]. In a consolidated effort, the collection was relocated in 2023 [8].

Genebanks are exposed to manifold endogenous and exogenous risks. However, a systematic and comparative assessment of their exposure, risks, and vulnerabilities is still missing. Also, risk management at genebanks—an emerging topic—has been barely studied scientifically. This study aims to fill these research gaps based on a selection comprising the world’s most important genebanks.

## 2. Objectives and Methodology

The objectives of this study were twofold: (1) to compare the most important genebanks worldwide with respect to their exposure to natural hazards and political and financial risks, and (2) based on this risk analysis, to develop risk management strategies for genebanks.

Based on these objectives, two methodological steps have been undertaken:Developing a risk analysis and risk management framework suitable for genebanks;Applying this framework to a selected, representative sample of the most important germplasm holdings worldwide.

### 2.1. Risk Analysis and Risk Management Framework

A methodology was developed following definitions, concepts, and frameworks for risk analysis and management used in the scientific literature of environmental hazard appraisals [9] and entrepreneurial risk management approaches [10], which were deemed the most suited for evaluating genebanks. The term *risk management* was used as defined by Wolke [10], involving four steps. These were refined and interpreted for this study as follows:


**Step 1: Risk identification**


Based on a literature review, expert interviews, and own assessments, the most important risks were classified as follows:Natural hazards (exogenous risks);Political risks (exogenous risks);Financial risks (exo- and endogenous risks).


**Step 2: Risk measurement and analysis**


The definition of risk as proposed by Dalezios (2017) [9] was used in this study:
risk=hazard×vulnerability×amount of elements at risk


The term *hazard* in the above formula is covered by the exposure assessment under Section 3 and is a main part of the study. For exogenous risks, a site-specific quantitative and comparative analysis of the sample of genebanks is provided for natural hazards and a country-specific analysis for political risks. Financing risks are discussed in qualitative and rather general terms due to a lack of data.

The term *vulnerability* of a genebank, i.e., “the extent to which an element at risk can withstand the impact of the hazard” [11] (p. 7), is strongly influenced by its conservation mandate (crops conserved and conservation methods used), the infrastructure (e.g., adherence to building codes), institutional organization, and on-site management. Assessing the vulnerability for each of the sampled genebanks was beyond the scope of this study; therefore, it is covered only in general terms (please note that the OECD [11] states, “Vulnerability … is even more difficult to quantify. … The scarcity and inconsistencies of vulnerability information often makes it the weakest link in a risk assessment” (pp. 7–8).).

The term *amount of elements at risk* comprises the buildings and facilities, personnel, and the germplasm collections of a given genebank.


**Step 3: Risk steering**


In this step, the strategies and instruments to control risks are discussed, especially risk prevention measures and—in case risks cannot be avoided or prevented—risk transfer solutions. Risk transfer means to transfer the financial consequences of a risk from the risk owner to a third party through mechanisms like insurance or funds [12].


**Step 4: Risk controlling**


This step deals with the question of how risks can be controlled to guarantee that they are properly addressed and managed. However, it is not a focus of this study, but it will be included in order to give a comprehensive overview on the management of risks inside genebanks.

Steps 3 and 4 will be combined under the term *risk management* and dealt with in Section 4.

### 2.2. Methodology for the Selection of Genebanks

This study aimed to cover the world’s most important ex situ holdings of plant genetic resources for food and agriculture (PGRFA) in terms of conserved accessions that are maintained by genebanks, including seed and field genebanks, as well as in vitro and cryopreservation facilities at international, regional, and national levels. Currently, only crop-specific rankings of germplasm collections are available, e.g., by the FAO [3] or through the global crop conservation strategies by the Crop Trust [13]), but s no global ranking of the 1750 genebanks worldwide with respect to their overall collection size in terms of number of accessions [14]. Although the number of accessions does not always correspond with the importance of the PGRA stored at a genebank, it is presently one of the least complex and, hence, most operational parameters for a ranking of genebanks and, therefore, used here. Unfortunately, there are still no good measurements for the total diversity that is included in a collection. 

To overcome above mentioned challenges, a sampling approach was developed that was orientated towards the organizational structure of genebanks, in particular differentiating between international and regional versus national genebanks. The classification is according to the WIEWS [15]. International genebanks comprise the CGIAR genebanks, the World Vegetable Center and the global safety duplication site SGSV. Regional genebanks have a mandate for conserving PGR in specific geographical world regions (e.g., SADC Plant Genetic Resources Centre (SRGB) in Zambia for Southern Africa). National genebanks refer to holdings at the country level. 

Key information for the sampling was compiled from two sources: the WIEWS database [15] and the FAO’s second report on *The State of the World’s Plant Genetic Resources for Food and Agriculture* [3] as well as the associated country reports. The data analysed in this study stems from the year 2020 and comprises accessions of plant genetic resources (including agricultural crops and its crop wild relatives) conserved under medium- and long-term storage [15]. FAO (2010) and the respective country reports allow identification and confirmation of the relevance of the sampled institutions. In a final step—after the genebanks have been sampled—the precise location of the selected institutions was validated via satellite images of DigitalGlobe/GeoEye.

The identification and selection of genebanks at international and regional levels proved to be without major challenges due to the limited number of institutions worldwide. In contrast to that, the relevant national genebanks were difficult to identify and assess due to the lack of concise and comparable information across the large number of countries, national institutions, and subsidiaries. In the following, the sampling approach of these genebanks is outlined, differentiating between international, regional, and national genebanks.


**Selection of international and regional genebanks**


International and regional genebanks typically aim for long-term conservation and facilitated access to accessions, predominantly of major food crop genepools that are in the public domain. Consequently, it was deemed most appropriate to comprise in the sample all 13 international genebanks and 8 regional genebanks. In addition, the safety duplication backup facility SGSV at Svalbard in Norway was included and categorized as an international genebank (resulting in 14 international genebanks). Being the world’s largest backup facility for seeds designed with the highest security standards: located at 130 m above sea level in a mountain with permafrost conditions and an additional cooling system bringing the seed storage temperature to minus 18 °C [16]. Electricity is provided by a public power plant or in case of power outage by generators [16]. Despite this, climate change is imposing new threats, e.g., in 2022, ice melted, and water entered the entrance section of the facility [17]. Nevertheless it can be regarded as a benchmark with respect to safety characteristics.


**Selection of national genebanks**


The selection was performed in different steps following specific criteria:Only state-managed and publicly funded national genebanks were included in the sample.The selection of national genebanks was performed via the identification of the most important agricultural countries using their gross production value for crops (GPV) and the cropping area. The latter was based on two FAO statistics: (1) the total area harvested and (2) the area of land used for agriculture. This approach followed the underlying rationale that a country that is an important agricultural producer is likely to possess a well-established agricultural (research) infrastructure, including facilities for the conservation of plant genetic resources. The first criterium, the GPV, reflects the economic value of the national agricultural sector, while the cropping area refers to the spatial importance of agriculture. The data were accessed via FAOSTAT [18]. Thirty-six countries were selected.After selection of the countries, the most important ex situ holdings for PGRFA at the national level had to be identified. For this step, the WIEWS database, the second report on *The State of the World’s Plant Genetic Resources for Food and Agriculture*, and relevant country reports were consulted, and an additional web research was conducted, among others, to confirm that the sample comprised only state-managed genebanks. In countries with a decentralized conservation system, more than one location was included in the sample.

The result of the above sampling approach for national genebanks was 58 national genebanks in the selected 36 countries, comprising a total of at least 3,857,013 accessions (Please note: For the Chinese duplication genebank, information about the collection size was not available. The Nottingham Arabidopsis Stock Center (NASC) was included in the sample due to its importance for research and development in plant genetics and breeding, despite the fact that the majority of the accessions is not PGRFA.), and is shown in Figure 1, which indicates that a relatively good geographical distribution was achieved by this sampling approach. Together with the selected international and regional genebanks, the sample consists of 80 genebanks in total (see Appendix B).

## 3. Hazard Assessment of Genebanks

### 3.1. Natural Hazards

The natural hazard assessment was done in cooperation with Munich Reinsurance Company (Munich Re), which provided the results of the location-specific natural hazard assessments of the 80 genebanks using their internal tool “Natural Hazards EditionThe tool allows single-risk assessments for 12 natural hazards and geospatial data analysis. It is based on the database NatCatService [19] and used by Munich Re worldwide for their single risk assessments. (Please note that natural hazard assessment tools are not publicly available. Therefore, presently there is no benchmark available to assess the quality of the “Natural Hazards Edition” tool. In general terms, OECD [11] states in this respect: “The need for better risk assessment data and tools therefore remains high” (p. 3).)

Twelve natural hazards—those considered the most relevant—were analysed and grouped into four categories:Geological hazards: earthquake, volcano;Hydrological hazards: tsunami, storm surge, river, and flash flood;Meteorological hazards: tropical cyclone, extratropical storm, tornado, hailstorm, and lightning;Climatological hazards: wildfire.

The hazards are represented as ordinal data in discrete categories following a multinomial distribution. Categories are defined individually for each hazard. The hazard assessment was conducted in three steps: (1) a single hazard assessment, done individually per hazard and aggregated across the sample of genebanks, (2) a hazard assessment per genebank location, and (3) the calculation of a global risk score per genebank location. Here, only the results of steps 2 and 3 are presented and discussed in detail.

Table 1 presents the results of step 2 exemplarily for international genebanks (for all genebanks, see Appendix A). From a risk management perspective, attention must be especially placed on locations with high to extreme exposure to individual hazards. These have considerable exposure to either one major natural hazard (e.g., earthquake for ICARDA in Lebanon and CIP in Peru) or to multiple hazards (e.g., volcano, flood, and tropical cyclones at IRRI, Philippines, or earthquake, volcano, hail, and lightning at ILRI in Ethiopia). However, because the hazard classes are not directly comparable across hazards, step 3 of the analysis is required for a comparative analysis of the hazard exposure across genebank locations.

For a comprehensive overview of a location’s exposure to natural hazards, Munich Re developed a weighted global risk index and a risk score for “ordinary commercial and industrial business” [20]. Because genebanks have similar risk characteristics as “ordinary commercial and industrial business”, the index and score can be used here. The global risk index and risk score build upon the hazard zones of the exposure assessment, loss expectations, and expert knowledge of Munich Re to weigh the hazards adequately [20]). As this is confidential information, Munich Re could not disclose it in detail [20]. Despite this limitation, the risk index and score are considered useful for this analysis, as it allows a quantitative comparison of the risk to natural hazards across locations. 

The global risk index is a quantitative value calculated as the sum of three individual risk indices that are considered globally the most important ones: the earthquake risk index (comprising earthquake, volcano, and tsunami risks), the storm risk index (comprising tropical cyclone, extratropical storm, hail, tornado, and lightning risks), and the flood risk index (comprising river flood, flash flood, and storm surge risks) [21]. The global risk index ranges from 0 (no risk) to 300 (extreme risk), whereby a risk index of 300 is of theoretical nature, reached only if all three hazard groups would have the maximum value. Therefore, values above 200 are considered highly unlikely in reality [21]. 

The results of the exposure analysis for the sample of genebanks are shown in Figure 2 (the natural hazard exposure assessment with the respective risk indices and scores are documented for all genebanks in the Appendix A). On average, the analysed genebanks have a risk index of 31.75 (arithmetic mean) and a median of 17. The lowest risk index has been calculated for the subsidiary of the international genebank ICARDA in Morocco (score = 5), and the highest index for the national genebank of the Philippines (PHL129) with a score of 152. Regionally, genebanks in Asia and Oceania are the most exposed, having a global risk index with an arithmetic mean of 57 and 54, respectively. The global risk index for the assessed genebanks on the African continent is the lowest, with an arithmetic mean of 14, followed by Europe with 21 and the American continent with 27. Yet, considerable differences across locations within a given continent exist.

Based on the global risk score, 35 genebank locations (44% of the 80 sampled genebanks) were classified with a low (one location) to medium risk (34 locations); 17 genebank locations (21%) have a high and 28 (35%) an extreme risk score. Most of the genebanks with an extreme risk score are located in Asia (14), followed by the Americas (6) and Europe (5). In percentage terms, two-thirds of the assessed Asian genebank locations are classified with an extreme global risk score. In Oceania, 33% of the genebank locations are rated with an extreme risk; in the Americas, 30%, and in Europe, 26%. The lowest share of institutions exposed to extreme risks can be found in Africa, where only 12% of the African genebank locations are classified with the highest risk score 4.

To conclude, from Figure 2, it is evident that the risk exposure varies spatially to a great extent.

Figure 3 shows the risk indices for the three hazard groups. It illustrates that the exposure to earthquakes is most relevant for genebanks in the Americas and, to a lower extent, for Asian and African locations. In Europa, flood risks are the most important, while Asian and Oceanian genebanks are exposed to all three hazard groups, amongst which storm is classified the highest, with extremely high risk indices.

Figure 4 gives a detailed analysis for the respective global risk indexes for international genebanks. Six genebanks are classified with the highest risk. Among these, the WorldVeg in Taiwan is the most exposed institution with a global risk index of 139, followed by IRRI in the Philippines (global risk index = 112) and CIP in Peru (global risk index = 78). In addition to Figure 4, three regional genebanks show an extreme high risk score. For the regional genebank CePaCT in Fiji, an overall risk index of 142 has been calculated, one of the highest exposed institutions of the entire group of sampled genebanks. Also, the regional genebanks CATIE in Costa Rica and NORDGEN in Sweden show a high risk, with an index of 49 and 42, respectively. The safety backup facility SVSG in Svalbard shows a comparable low exposure to natural hazards (with a risk index of 9). As its exposure is mainly driven by storm events, this is of minor relevance because the building’s infrastructure is mostly located underground and, therefore, has a very low vulnerability to storm.

### 3.2. Political Risks

Due to the complex nature of political risks, they have been analysed at the country level using two indicators: the Worldwide Governance Indicator (WGI), published by the World Bank [22], and the Fragile States Index (FSI), published by The Fund for Peace [23]. To account for possible between-year differences, the six-year average from 2015 to 2020 was calculated for each of the indicators.

The WGI covers six dimensions of governance: (1) voice and accountability, (2) political stability and absence of violence, (3) governance effectiveness, (4) regulatory quality, (5) rule of law, and (6) control of corruption [24]. The indicators are expressed on a scale from −2.5 (weak governance) to +2.5 (strong governance), centred around 0. For this study, all WGI dimensions were considered to be of relevance when assessing political risks for genebanks. Consequently, a total WGI score was calculated as the unweighted average across the six dimensions, also ranging from −2.5 to +2.5.

The FSI by the Fund for Peace assesses the risk of a state failure based on a total score ranging from 1 (low risk) to 120 (high risk). It is calculated as the unweighted sum of twelve indicators covering five aspects of political stability: cohesion, economic, political, social, and cross-cutting indicators [23]. For this study, the total score was used.

The results are illustrated in Figure 5 for the WGI indicator and in Figure 6 for the FSI indicator, using score colours for all 47 countries where the sampled genebanks are located. This gives a visual impression pointing to a good concurrency of the two indicators, despite the methodological differences of the indicators and general methodological challenges associated with quantifying political risks. The concurrency is further confirmed by Figure 7, providing a ranking of the sampled countries for these two indicators. The countries most at risk are ranked highest (left lower corner in Figure 7), rising to the most stable nations (right upper corner). From this, it becomes apparent that the two indicators coincide for the majority of the countries relatively well, i.e., graphically, the countries are close to the plotted line through the origin.

Overall, African and some Asian countries were assessed with a high political instability, whereas general European (except for Ukraine and the Russian Federation) and North American countries, particularly Canada and USA, show a high political stability. Notably, some important international genebanks are located in countries assessed with a high political risk: IITA in Nigeria, ILRI in Ethiopia, ICARDA in Lebanon, ICRAF in Kenya, and AfricaRice in Côte d’Ivoire.

### 3.3. Financial Risks

Financing constraints, implying an insufficient level of funding and non-reliability of funds, are a major threat to sustainable PGR conservation, as they adversely affect genebank operations, functioning, and risk management. Based on the literature review and expert interviews, a tentative qualitative exposure ranking (highest to lowest financial risk) can be given as follows:National genebanks in developing countries;National genebanks in emerging economies and some developed countries with decentralized structures (decentralized structures seem to be more vulnerable, as their funding often comes from different sources, e.g., besides central, also regional governments) and weak national coordination;International CGIAR genebanks;National genebanks in developed countries with centralized management or decentralized structures with strong national coordination.

In general, information on the actual and required budget per genebank, as well as on the nature and provenance of funds, is scarce. Therefore, a thorough analysis of the financial situation of the sampled genebanks could not be carried out. However, it can be noted that the international genebanks have—in particular, through the professional work of the Crop Trust operating the Crop Diversity Endowment Fund—a stable financial backbone, whereas the situation of national genebanks is very diverse. It depends on the organizational and management structure (e.g., centralized vs. decentralized), the overall state budget, and the priority-setting of national governments.

## 4. Risk Management at Genebanks

After the above-described assessment of risks and hazards, in a further step, strategies and instruments to manage risks should be addressed. There is a global tendency toward an increased exposure to hazards—especially to natural hazards, largely as a consequence of climate change and with respect to political risks due to increased international instability. Therefore, risk steering and controlling are becoming more and more critically important and should become a core activity for secured PGR conservation at genebanks. Hence, the development of criteria and standards for an effective and efficient risk management, as well as respective staff training, will be essential. Two areas are of particular importance: (1) risk prevention and mitigation and (2) risk transfer.

### 4.1. Specific Risk Prevention and Mitigation Strategies for Genebanks

Risk prevention aims at avoiding emerging and existing risks from materializing, while the objective of risk mitigation is to reduce the impact of hazardous events [25]. Hence, both have a prospective character. For managing specifically exogenous risks at genebanks, two important strategies can be identified: (1) increasing resilience of infrastructure and (2) safety duplication of accessions [6,26,27,28].


**Increasing resilience of infrastructure**


Increasing the resilience of infrastructure, e.g., buildings, technical facilities, and IT infrastructure, mitigates the impact of natural hazards, electricity outages, and malfunctioning of technical devices. Based on the literature review, expert interviews, and own assessments, Table 2 has been compiled, which summarizes the most important risk control measures.

For the twelve assessed natural hazards, the infrastructural standards are suggested according to Table 3. Here, only the genebanks that are part of the sample and have been assessed with the highest exposure to the respective natural hazard are mentioned. Especially for these genebanks, the standards are of high priority. However, the suggested measures are obviously not limited to these genebanks only.

It should be highlighted that most of the above-mentioned risk control measures apply to seed genebanks as well as in vitro and cryopreservation facilities. In addition, for in vitro conservation, a high emphasis needs to be put on the control of technical installations and equipment, as specific temperature and light requirements have to be met. A. W. Ebert (personal communication, 25 July 2022) [39] points to the risk of high temperatures above 40 °C; if the air conditioning is defective, this could put the entire in vitro collection in danger. By contrast, for field genebanks, specific risk control measures are needed, as they are exposed to additional hazards (e.g., pests, diseases, theft and animal damages, drought, and flooding). Most of these are difficult to control in the field, so the location of the genebank and protective infrastructure like fences, hail nets, and irrigation are crucial [33,39,47].

If a genebank will be constructed new or renovated it is recommended that a thorough exposure and vulnerability assessment is conducted beforehand and that the respective building codes are applied. As part of such an assessment, the risk of pathogen pressure at a specific location should also be considered, as this can be mitigated substantially through choosing a genebank’s location; for instance, in no cropping areas or dry areas to lower the disease pressure on seeds.


**Safety duplication of orthodox seeds**


Maintaining safety duplicates of accessions at two or even three different locations in another country and possibly on another continent [47] is an important risk management strategy in ex situ conservation. It has proven to be effective already in the past to restore lost accessions or even entire collections (e.g., of ICARDA in Syria). Therefore, it is promoted by stakeholders and researchers and widely applied at national and international genebanks—yet at a varying level [3,6,26,28].

As per the FAO Genebank Standards, most genebanks should have safety duplication arrangements with one or more institutions, including international, regional, and national genebanks, as well as the SGSV [3]. Preferably so-called black box agreements are applied, meaning that the recipient institution conserves the duplicate but has neither rights over it nor further obligations (i.e., is not responsible for viability testing and is not allowed to regenerate, use, or distribute the material if not authorized by the depositor) [13,28].

Genebanks have adopted different strategies: either a system of duplicates (e.g., the IPK in Germany where accessions are safety duplicated only at SGSV) [48] or of triplicates (e.g., the Dutch CGN where accessions are safety duplicated at another national genebank and at SGSV) [3,33,35,39]. Moreover, duplication at another active genebank is a valid approach. This means that the collection is not only stored for conservation but also actively used.

These risk mitigation strategies of safety duplicates come at a certain cost, requiring substantial financial resources, sufficient storage capacities, legal and institutional agreements, and a good documentation and information system [28,33,35,49]. This points to an important implication for the global conservation system: the decision of what material should be safety duplicated requires a prioritization (although it is desirable that all or at least the majority of accessions of a collection are safety duplicated, this is often not possible because of financial constraints). This implies a complex value judgment based on a thorough assessment of the importance and value of individual accessions. At the German genebank IPK and at Plant Gene Resources of Canada, this prioritization is purely governed by logistics, i.e., only recently multiplied material is sent to SGSV [33], but over time, the whole collection will be duplicated there.

Another important limitation is the high level of unintended duplicates within collections. The FAO (2010) estimates that only between 25 to 30% of the accessions in ex situ collections are unique. Therefore, despite recognizing that the ultimate goal should be safety duplicating the whole collection, it is recommended to establish guidelines on how to prioritize accessions for safety duplications based on common principles but flexible according to the respective national context and financial resources. An interesting example is Canada, where results of molecular marker analysis such as accession distinctness are taken into account when prioritizing [49].

With the opening of SGSV in 2008, an important milestone with respect to safety duplication and backup of accessions was achieved. This unique storage facility with a capacity of 4.5 million accessions has the highest safety standards and, therefore, is the world’s most important safety backup facility. In February 2023, more than 1.2 million seed samples of more than 5000 plant species coming from 98 institutions in 76 countries have been stored there [16]. The largest numbers of accessions stored are varieties of rice and wheat (each >150,000), followed by barley (close to 80,000), sorghum (>50,000), *Phaseolus* bean species (>40,000), maize (>35,000), cowpea (>30,000), and soybean (>25,000) [16]. About two-thirds of the presently deposited accessions are from the international genebanks. Among national genebanks, the USA, Germany, Canada, and the Netherlands are the main depositors, while for regional genebanks, NORDGEN is the main depositor [16].

Despite the above-mentioned positive development and the considerable progress made during the last two decades with respect to the percentage of safety-duplicated material, important parts of ex situ collections still “remain inadequately safety duplicated” ([3] p. 87). This applies especially to crops that cannot be maintained as seeds, i.e., vegetatively propagated crops or recalcitrant seeds (see below) and to national seed genebanks in some developing countries due to scarce financial resources [3].

The rate of accessions being safety duplicated is in general higher in CGIAR genebanks than at most national institutions [3,16]. This achievement is especially triggered by the Crop Trust, which links its financial support for genebanks to performance targets, including the rate of safety duplications [50]. However, even there, the external reviews conducted between 2017 to 2021 criticised a lack of sufficient safety duplications at some of the CGIAR genebanks, e.g., the ICRAF in Kenya and ICARDA in Morocco and Lebanon [51].


**Safety duplications of vegetatively propagated crops and non-orthodox seeds**


Vegetatively propagated crops and non-orthodox seeds, comprising intermediate and recalcitrant seeds, are predominantly maintained in field genebanks. The accessions have a high vulnerability, as risk mitigation through infrastructural means is limited to irrigation facilities, hail nets, and fences for their protection. This, however, is much less effective in comparison to seed genebanks [47]. Notwithstanding this constraint, the level of safety duplication is considerably lower compared to orthodox seeds. Therefore, from a risk management perspective, more efforts to secure field genebank accessions are necessary.

For risk mitigation, there are two main ways of safety duplicating these crops, as detailed by FAO (2014) [47]:Duplication of field collections at another location (not exposed to the similar risks as the original field genebank);Duplication of field genebank accessions under alternative conservation methods, such as in vitro conservation and cryopreservation.

These approaches are especially important, as presently, there is no global backup conservation facility available for field collections, i.e., conserved in vitro or cryopreserved, similar to the SGSV for seed collections.

In vitro conservation and cryopreservation are less susceptible to natural hazards compared to field genebanks, which are directly exposed to environmental risks. In vitro and cryostorage reduce the vulnerability to natural hazards by moving the collection from an exposed outside location to controlled inside conditions. The level of vulnerability, then, depends predominantly on the building infrastructure, the technical facilities securing a controlled environment, and the operating staff. Moreover, a building can be better secured against human-related damages, such as theft, vandalism, and political risks. Furthermore, the accessions maintained in field genebanks are also directly and continuously exposed to pests and diseases. Especially infections with viruses cause severe problems to the genebank, as these might impede their distribution and exchange with other genebanks because of quarantine regulations. During the process of preparing materials for in vitro and cryopreservation storage, viruses can be eliminated through specific treatments; thus, cleaned in vitro/cryopreserved samples can be safely exchanged.

However, there are two major drawbacks to these methods:The costs for setting up and introducing material to in vitro and cryopreservation are considerably higher than for field genebanks. However, in the case of cryopreservation, once the system is established, its running costs are relatively low [52]. (Note that in vitro conservation is not suitable for mid-term and long-term conservation. But in vitro is important in connection with cryopreservation, as the plant tissue material first has to be prepared in vitro before it can be stored in liquid nitrogen [39].)Both alternative methods require a high level of training to manage conservation appropriately [1,52].

According to Panis et al. (2020) [52], duplicating field genebank accessions in in vitro or cryopreservation at another location is recommended. Cryopreservation is especially suitable for secure long-term conservation, as it requires regeneration only after several hundred years [52]. However, if financial resources are scarce, duplicating the field genebank collection at another distant field location might be an appropriate alternative [28].

### 4.2. Risk Transfer Strategies

Risk transfer is a key strategy to manage risks and is particularly relevant if risks cannot be prevented. It transfers the financial consequences of a risk from the risk owner to a third party through different mechanisms like insurance schemes or funds [12]. Risk transfer solutions are common in different economic sectors, among others in agricultural production. However, in agrobiodiversity conservation, they are currently hardly applied.


**Insurance solutions**


The most common and widespread risk transfer solutions are insurance coverages [12]. Insurances are financial agreements to transfer defined risks to a third party, the insurer, against the payment of agreed monetary terms (premium)—and this should be done before the risk materializes [12]. The advantages of insurance covers are that there is a legal entitlement for indemnification and that the insurance company normally carries out a risk assessment, including identifying and requesting risk prevention measures.

There is only limited information available if and to what extent genebank assets are covered by insurance schemes. Based on the literature review [53,54,55] and expert interviews (F. Begemann (personal communication, 11 August 2022); A. W. Ebert (personal communication, 25 July 2022); U. Lohwasser (personal communication, 15 July 2022); T. van Hintum (personal communication, 28 July 2022) [33,35,39,56]), it can be concluded that in most countries, genebanks are not insured. This is due to the fact that most genebanks are in public ownership, and in case of an emergency, the state is supposed to bail out and rebuild facilities and infrastructure. This premise, however, seems a questionable strategy for the future, considering three trends:Natural hazards will increase in frequency and intensity due to climate change, augmenting the exposure of genebanks and other infrastructure [57].An increased concentration of ex situ conservation structures at the country and institutional levels, as well as increased numbers of accessions stored, will increase the values at risk in future [3,6].Financial constraints of states and decreasing political support for PGR conservation [6]; with the recent increase in interest rates in important economies like the USA and Europe, governmental budget limitations are likely to become more important while financing debts (e.g., as necessary in the aftermath of disasters) will probably become more difficult in future.

Based on these considerations, it is recommendable to include insurance schemes for genebanks as a complementary risk management strategy in future. The relevant issues and challenges in this process are briefly discussed below.

In most cases, insurance coverages are offered by nationally approved insurance companies. Genebanks can access this insurance capacity. Additionally, for genebanks in developing countries, risk pooling insurance instruments for natural hazards might also be relevant. Interesting examples at the regional level are the Caribbean Catastrophe Risk Insurance Facility (CCRIF) and the African Risk Capacity (ARC) (for further information, see [58,59,60]). Internationally organized facilities like the CGIAR genebanks could potentially also look for an umbrella cover for all its genebanks. This would primarily be offered by globally organized insurance and reinsurance companies.

The insured perils in insurance contracts are usually fire and explosion as well as all major natural hazards (e.g., earthquake, volcano, storm, hail, flood). However, locations with a very high flood exposure, e.g., close to rivers and creeks, might not be eligible for flood coverage. It is important to note that standard exclusions in insurance policies are war, terrorism, and radioactive contamination. As genebanks have been damaged or destroyed by war acts in the past, this is certainly an important limitation.

The most critical insured assets of genebanks are:Buildings and storage rooms;Technical facilities and equipment (e.g., refrigerated storage facilities, control units, alarm systems, laboratory);Germplasm collections.

From a risk management point of view, it is recommendable to insure all asset classes, but it is also possible to select specific ones, e.g., only buildings and storage facilities or germplasm collections. Insured assets are covered for physical loss, damage, or destruction caused by an insured peril.

An essential step in structuring an insurance contract is the valuation of the assets. Based on the valuation, the sums insured per insured asset are defined, and these are the basis for any indemnity paid after a loss event. For the genebank infrastructure (buildings and technical facilities), this is relatively easy to estimate using market or replacement values. However, valuing PGR collections is challenging because PGR accessions are non-tradable items and, hence, do not possess a market or replacement value. For the valuation of PGR collections, two approaches are most suitable:**Replacement value:** In this approach, the cost to replace and rehabilitate any collection lost is determined. In the case of accessions, a replacement is only possible if the accessions are stored as safety duplicates elsewhere and are accessible and viable. Such a replacement exercise was undertaken in the case of the CGIAR genebank ICARDA in Aleppo, whose collections have been restored in Morocco and Lebanon since 2016 using backed-up accessions at other genebanks and SGSV [4,5]. The costs of this operation are, however, not publicly available at present. Another reference is the Dutch genebank, where replacement costs have been recently estimated at €25 to €30 million overall [35]. This would result in a value of €1040 to €1250 per accession. As replacement operations are complex and costly, the respective figures are on the high side.**Costs of conserving accessions:** Using the costs of conserving accessions as an approximation for estimating the value of germplasm collections is an indirect approach. The advantage is that costs are relatively easy to establish [61]. Thereby, the costs of conservation in perpetuity should be used, as they focus on the long-term preservation of plant genetic material. As conservation costs are reasonable, this approach results in a relatively low level of valuation. Koo et al. (2003) [62] also used this approach and collected crop-specific in perpetuity costs at five international CGIAR genebanks. These data—even though dating back to the late 1990s and early 2000s—are the best available. Table 4 compiles the data of Koo et al. (2003) [62] and derives from these present values. These authors worked with different interest rates (2%, 4%, and 6%), which have a considerable impact on the value estimation (cf. Table 4).

Additionally, Rabenau (2018) [64] collected in perpetuity cost data at the German national genebank IKP in Gatersleben based on the methodology of Koo et al. (2003) [62] (cf. Table 5).

From both tables, it is obvious that the estimated value per accession varies considerably between crops. Also, the way regeneration is conducted influences the results, e.g., in Table 4, with and without initial regeneration of wheat and maize, and in Table 5, regeneration in open fields vs. greenhouse regeneration for soybean and chickpea.

In summary, to determine the actual value of a specific germplasm collection, it would be best to establish own cost data using the methodology of Koo et al. (2003) [62] and, based on these, to estimate the total value of the collection, e.g., for insurance purposes to establish the insured value of the collection. To account for particular features of the collection, e.g., share of unique accessions, respective loading factors might be used.


**Funds**


Funds—defined as a pool of money that is allocated for a specific purpose [54]—are another important risk transfer instrument. They can be financed through fees, donations, or financial resources granted by the state and are administered either by state institutions, self-governed bodies, or financial institutions [65,66]. Most funds are designed nationally, but also a supranational or even global scope is possible. Funds are often set up to respond to natural hazards, but any other peril, e.g., nuclear risks or terrorism, can also be covered, depending on the objective of the funders.

Similar to insurance coverages, funds allow the rapid mobilization of financial resources, circumventing time-consuming approval procedures and negotiations for accessing other financing sources after a disaster has occurred [67]. However, as funds are rather difficult and complex to set up, they are usually designed for circumstances where insurance coverages are neither available nor cost-effective.

For designing and implementing a fund, the most critical issues to be addressed include:The geographical scope of the fund;The assets to be covered;Value of the respective assets (as described for insurance solutions);The perils covered;Size and capacity of the fund;Financing of the fund: e.g., either through fees/premiums paid by the participating genebanks, deposits of donors (state or private), or a mixture of both.

In this context, the Global Crop Diversity Trust has taken an active role in establishing in 2021, jointly with the Secretariat of the ITPGRFA, the fund named Emergency Reserve for Genebanks [68]. Although targeting national genebanks in developing countries, the fund is open for national and international seed and field genebank collections, provided a substantial financial need can be demonstrated. Different interventions can be financed, e.g., repairing technical facilities, relocations of collections, or safety duplications of threatened unique accessions [50,69]. This fund, in contrast with the Endowment Fund, uses the financial resources of donors allocated to the fund directly. Therefore, if financial resources are spent, additional money has to be acquired. At present, a target for the financial volume of the fund has not been specified, as the number and scale of future requests are difficult to predict. Notwithstanding the presently limited volume, such a fund is considerable progress towards an improved risk transfer for genebanks.

In addition, other national and regional funds covering the aftermath of natural disasters exist. Two interesting examples that might be of use for genebanks are the Mexican fund FONDEN at the national level and the European Union Solidarity Fund (EUSF) at the regional level (for further information, see European Commission (2019) [70] and World Bank (2012) [71]). FONDEN, established in the late 1990s, is designed to support the quick rehabilitation of public infrastructure after adverse natural events [71]. Exploring the integration of genebanks into these national and regional funds is recommended.

It can be concluded that funds solely or in combination with insurance schemes could be a feasible risk transfer mechanism for genebanks that is worthwhile exploring and developing further.

## 5. Findings and Conclusions

This study provides a comprehensive risk analysis and risk management framework for genebanks worldwide. Its main findings and recommendations are summarized as follows:The natural hazard exposure analysis for a sample of 80 international, regional, and national genebanks, covering 65% of the world’s accessions, shows that risk exposure is highly location-specific and varies considerably between the analysed genebanks. Overall, 35 genebanks (44% of the sampled institutions) show a low to medium risk, while 17 genebanks (21%) present a high and 28 (35%) an extreme risk. Most of the extremely exposed genebanks are located in the Asia-Pacific region (Philippines, Fiji, Taiwan, Japan, and Bangladesh) as well as in South America (Peru) and Europe (UK, Germany, and Poland) (see Appendix C and Table A3). On the other hand, genebanks in Africa tend to have relatively low exposure to natural hazards. Among the international and regional genebanks, the most exposed are CePaCT in Fiji, WorldVeg in Taiwan, IRRI in the Philippines, and CIP in Peru. In contrast, SGSV, the global backup storage facility of safety duplicates, shows a relatively low risk profile, being only exposed to extratropical storms and rising temperatures affecting the permafrost. As the storage rooms are located underground, the vulnerability of the facility towards storm can be assessed as very low.These findings entail the following consequences for risk management:A location- and institution-specific risk assessment is indispensable to define and carry out appropriate risk prevention methods using two main strategies: (1) adequate infrastructural measures like natural hazard-resistant building codes, storage facilities at higher levels (flood prevention), emergency backup generators, and alarm systems; and (2) safety duplication of accessions at another location. Both strategies can be implemented without major obstacles in the conservation of orthodox species but are more complicated when conserving clonal and recalcitrant species in field genebanks.Risk transfer solutions like insurance coverages and funds, at present hardly implemented at genebanks, should be considered when developing holistic risk management strategies of genebanks. An important step in this direction was taken by the Global Crop Diversity Trust with the set-up of the Emergency Reserve Fund in 2021. Prices of these solutions vary significantly in line with the site-specific risk exposure.Vulnerability is very site-specific, depending mainly on the quality of infrastructure and risk prevention measures in place. Furthermore, it differs according to the specific conservation methods. Conservation in seed genebanks is the most resilient method compared to in vitro and cryopreservation, as the latter ones imply high technical and technological requirements. Field genebanks have a distinct risk profile and have, among the common conservation methods, the highest vulnerability with respect to natural hazards as well as pest and disease incidences.Assessing the exposure to political risks is challenging due to the complex nature of political risks. Using the two international indicators, the WGI by the World Bank and FSI by The Fund for Peace, this study identified considerable differences in the political stability of countries. Among the most exposed countries of the sample, predominantly located in Africa and Asia, are countries hosting important international genebanks. From a risk management perspective, it would be essential to establish a centralized monitoring system for political risks (e.g., at FAO or the Crop Trust) to be able to take safety measures proactively and on time.The insufficient level of financing has widely been acknowledged as a key limiting factor for genebanks. Yet, information on the actual and required budget per genebank, as well as on the nature and provenance of funds, is scarce and difficult to obtain. More research is necessary; it is recommended to include in the FAO country reports a section about necessary financial resources. In addition, it can be noted that the international genebanks have—in particular through the professional work of the Crop Trust establishing the Crop Diversity Endowment Fund—a stable financial backbone. In contrast, the situation at national genebanks is very diverse. It depends on the organizational structure (centralized vs. decentralized), the overall state budget, and the priority-setting of national governments.

In summary, progress has been made in the last few years to mainstream and strengthen quality and risk management, in particular at international genebanks, with the Crop Trust being a driving force. The CGIAR genebanks, as well as some national genebanks (e.g., in Germany, the Netherlands, the USA, and Canada), are more advanced regarding risk management. However, at many national genebanks, considerable scope for improvement remains. Therefore, a location-specific hazard and vulnerability assessment is recommended in order to define appropriate risk prevention measures and risk management strategies at the genebank level. Any progress in this respect can only be achieved with adequate human and financial resources as well as political support.

## Figures and Tables

**Figure 1 plants-12-02874-f001:**
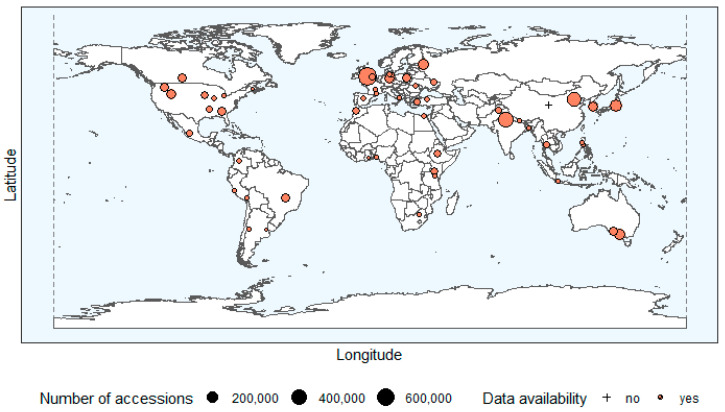
World map of sampled national genebanks and their collection size (circle size corresponds to number of accessions per genebank). If collection size was not available, it is marked with +. Source: adapted from WIEWS (2020) [15] and FAO country reports.

**Figure 2 plants-12-02874-f002:**
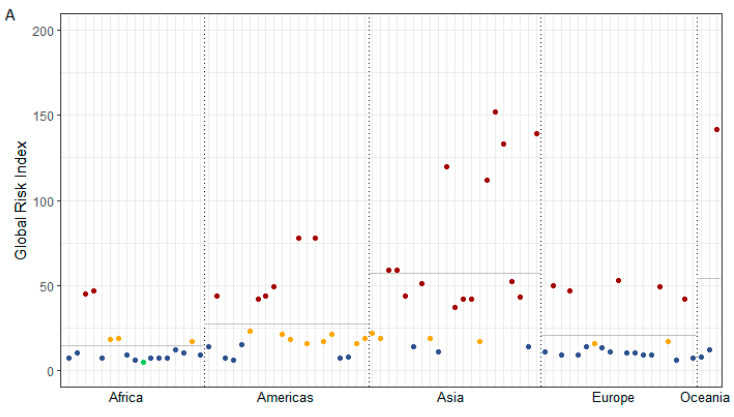
Natural hazard exposure analysis per individual genebank. (**A**) Global risk index (ranging from 0 (no risk) to 300 (extreme risk)), grouped by continent and coloured by global risk score (green = 1 (low), blue = 2 (medium), yellow = 3 (high), red = 4 (extreme)). Horizontal line means continental arithmetic mean of sample. (**B**) World map with global risk score: green = risk score 1 (low), blue = risk score 2 (medium), yellow = risk score 3 (high), red = risk score 4 (extreme). Source: adapted from Munich Re (2022) [19].

**Figure 3 plants-12-02874-f003:**
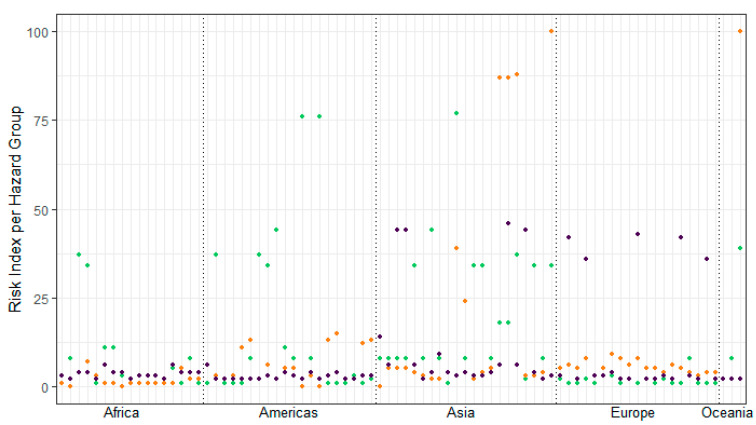
Disaggregated risk index (per hazard group, ranging from 0 (no risk) to 100 (extreme risk)) for sampled genebanks, grouped per continent, and coloured by risk group (green = risk index earthquake, orange = risk index storm, purple = risk index flood). Source: adapted from Munich Re (2022) [19].

**Figure 4 plants-12-02874-f004:**
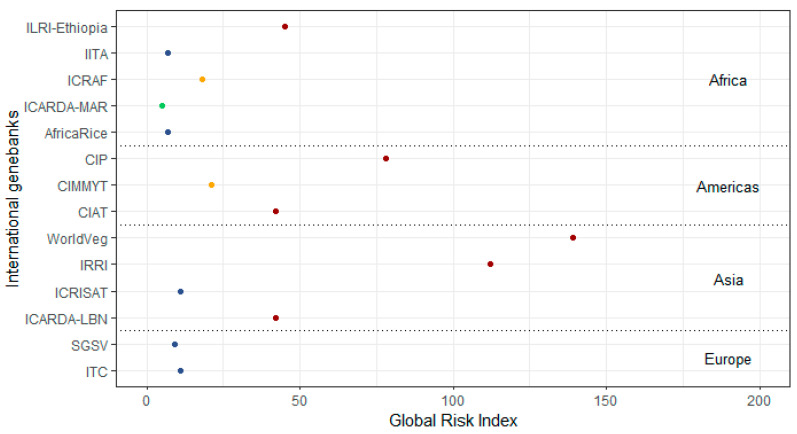
Global risk index for international genebanks, grouped per continent and coloured by global risk score (green = 1 (low), blue = 2 (medium), yellow = 3 (high), red = 4 (extreme)). Source: adapted from Munich Re (2022) [19].

**Figure 5 plants-12-02874-f005:**
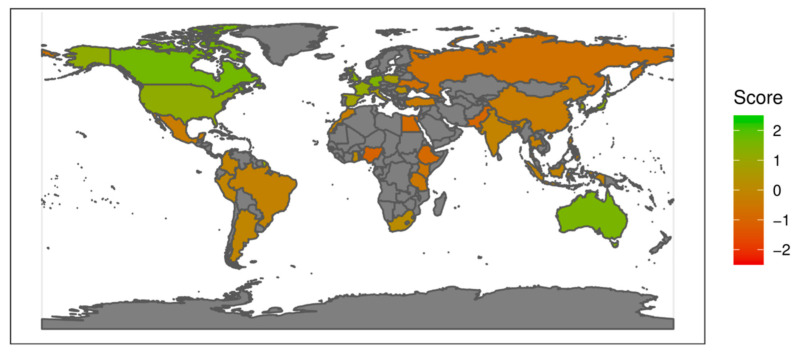
Mean WGI score: Map of all countries comprising the study’s genebank sample. Coloured according to the country’s average WGI score (six-year average 2015–2020, averaged across the six WGI dimensions) with a scale of −2.5 (low governance) to +2.5 (high governance); in grey = countries not in the sample. Source: adapted from World Bank (n.d.) [22].

**Figure 6 plants-12-02874-f006:**
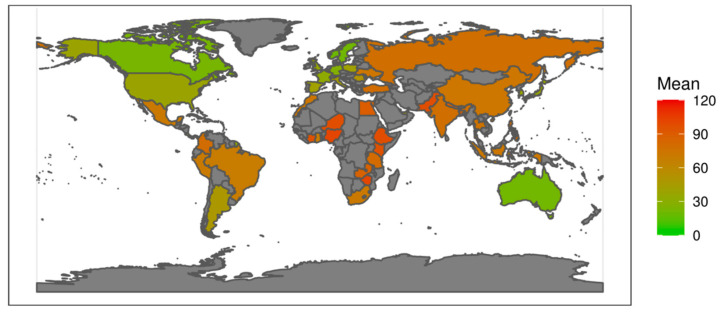
Average FSI total score: Map of all countries comprising the study’s genebank sample. Coloured according to the country’s average FSI total score (six-year average 2015–2020, unweighted sum of the 10 FSI dimensions) with a scale of 0 (low political risk) to 120 (high political risk); in grey = countries not in the sample. Source: adapted from The Fund for Peace (n.d.) [23].

**Figure 7 plants-12-02874-f007:**
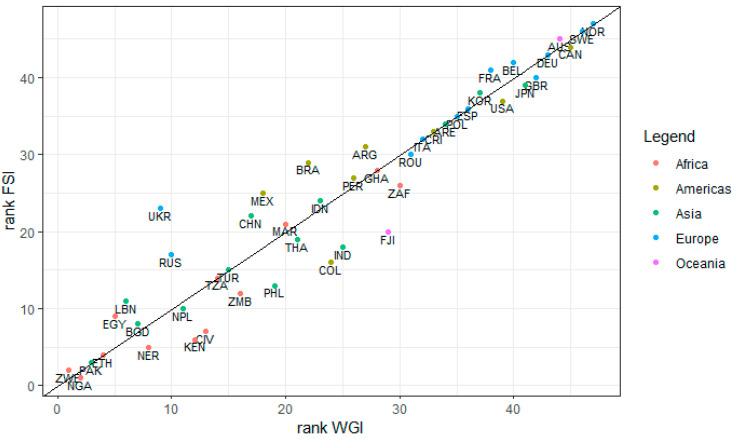
Comparison of the country ranking for political risks by average WGI score and FSI total score for all countries in the study hosting international, regional, and national genebanks, coloured by continent (with 1st rank = high political risk to 49th rank = lowest political risk of the sample). x-axis = country ranking according to the mean WGI score, calculated as the six-year average 2015–2020, averaged across the six WGI dimensions. y-axis = country ranking according to the FSI total score, calculated as the six-year average 2015–2020, unweighted sum of the 12 FSI dimensions. Source: own representation based on data provided by World Bank (n.d.) [22] and The Fund for Peace (n.d.) [23]. Country abbreviations: ARE = United Arab Emirates, ARG = Argentina, AUS = Australia, BEL = Belgium, BGD = Bangladesh, BRA = Brazil, CAN = Canada, CHN = China, CIV = Côte d’Ivoire, COL = Colombia, CRI = Costa Rica, DEU = Germany, EGY = Egypt, ESP = Spain, ETH = Ethiopia, FJI = Fiji, FRA = France, GBR = United Kingdom and Northern Ireland, GHA = Ghana, IDN = Indonesia, IND = India, JPN = Japan, KEN = Kenya, KOR = Republic of Korea, LBN = Lebanon, MAR = Morocco, MEX = Mexico, NER = Niger, NGA = Nigeria, NOR = Norway, NPL = Nepal, PAK = Pakistan, PER = Peru, PHL = Philippines, POL = Poland, POU = Romania, RUS = Russian Federation, SWE = Sweden, THA = Thailand, TUR = Türkiye, UKR = Ukraine, USA = United States of America, ZAF = South Africa, ZMB = Zambia, ZWE = Zimbabwe. Please note: Taiwan (hosting the international genebank WorldVeg) needed to be excluded for the analysis because it is only in the indicator WGI and not in FSI.

**Table 1 plants-12-02874-t001:** Natural hazard exposure assessment for international genebanks.

WIEWS Institutional Code	Organization Acronym	Country	Earthquake ^1^	Volcano	Tsunami	Storm Surge	Riverflood	Flash Flood	Tropical Cyclone	Extratropical Storm	Tornado	Hailstorm	Lightning	Wildfire
BEL084	ITC	Belgium	1	−1	−1	−1	0	3	−1	2	3 ^2^	2	2	1
CIV033	Africa Rice	Côte d’Ivoire	0	−1	−1	−1	0	3	−1	−1	1	2	4	2
COL003	CIAT	Colombia	3	2	−1	−1	0	2	−1	−1	1	3	4	1
ETH013	ILRI	Ethiopia	3	2	−1	−1	0	4	−1	−1	1	5	5	−1
IND002	ICRISAT	India	0	−1	−1	−1	0	4	0	−1	2	2	3	2
KEN056	ICRAF	Kenya	2	2	−1	−1	0	5	−1	−1	1	3	2	−1
LBN002	ICARDA	Lebanon	3	−1	−1	−1	0	3	−1	0	2	4	2	3
MARNA	ICARDA	Morocco	0	−1	−1	−1	0	3	−1	1	1	1	2	−1
MEX002	CIMMYT	Mexico	2	2	−1	−1	0	4	0	−1	1	5	4	1
NGA039	IITA	Nigeria	0	−1	−1	−1	0	2	−1	−1	1	2	4	3
PER001	CIP	Peru	4	−1	−1	−1	0	2	−1	−1	1	1	1	−1
PHL001	IRRI	Philippines	2	3	−1	−1	0	5	4	−1	1	2	4	1
TWN001	World Veg	Taiwan, Province of China	3	−1	−1	−1	0	3	5	0	1	2	2	2
NOR051	SGSV	Norway	0	−1	−1	−1	0	1	−1	3	1	1	/	−1

^1^ Note: exposure class varies between hazards. In general, the higher the number, the higher the exposure, with 5 presenting the highest and 1 the lowest exposure class. Minus 1 means no exposure to the hazard. For details, see Appendix C. ^2^ Highest exposure class per hazard in dark red, second-highest exposure class per hazard in light red. Source: adapted from Munich Re (2022) [19].

**Table 2 plants-12-02874-t002:** Infrastructural risk control measures for each of the risk sources.

Risk Source	Risk Control Measure
Natural hazards	Building codes, standards, and practices (e.g., resistance to earthquake, strong wind, and snow load). These are being regulated normally at the national level.
Outage or malfunctioning of technical facilities	Alarm systems (e.g., for open doors, sudden changes in light, temperature, and humidity); early fire, gas, smoke, or water detection; backup equipment or additional rooms available and ready; essential spare parts in storage; qualified staff for repairs or external standby repair services
Fire (ignition point inside the facility)	Detection and mitigation devices: Smoke and fire detection; sprinkler systems; fire extinguishers. Construction measures: fire walls; fire isolation doors; separated compartments; sufficient separation between buildings. Organizational aspects: coordination with external firefighting services like local fire brigades
Fire (ignition point outside the facility, e.g., wildfires)	Fire breaks; fuel load control in the vicinity of the genebank
Power supply cut-offs	Second power line, emergency power generator (for storage rooms, monitoring devices, essential lighting, etc.), lightning rods and deflectors
Theft, vandalism, and terrorism	Alarm systems, locks, surveillance cameras, and sensors to impede the entry of unauthorized people (in addition to security surveillance)
Cyber-attacks to IT ^1^	High cybersecurity standards

^1^ IT risks are currently underestimated but are expected to become increasingly important in future due to increased digitalization at the genebank level. Source: adapted from CGIAR Genebank Platform (2020) [26], Crop Genebank Knowledge Base (n.d.) [27], Fu (2017) [6], and expert interviews.

**Table 3 plants-12-02874-t003:** Infrastructural risk control measures suggested for genebanks with highest exposure, per natural hazard.

Natural Hazard	Risk Control Measures	Examples of Exposed Genebanks ^1^
Earthquakes	Earthquake-proof infrastructure [26]	- international genebank CIP, Peru (PER001) - national genebank of Peru (PER066) in Lima and of Japan (JPN183)
Volcanoes	Strengthened roofs and walls, use of shutters on openings and non-flammable materials, fix buildings to foundation, etc. [29]	- international genebank IRRI, Philippines (PHL001) - regional genebank CATIE, Costa Rica (CRI134/CRI142/CRI085) - national genebank of Indonesia (IDN179) and of the Philippines (PHL129)
Tsunami	Tsunami-resistant structures [30]	- international genebank CePaCT, Fiji (FJI049)
Storm surge	Storm surge gates, flood barriers, floor plans for a quick water outflow, shelving above ground level [26,31]	- regional genebank NORDGEN, Sweden (SWE054) - location Poel of national genebank of Germany (DEU271)
River flood	Flood barriers, dikes, spurs, etc., floor plans for a quick water outflow, shelving above ground level, water-proof ink and bags [26,32,33,34,35]	- national genebanks of Bangladesh (BGD002, BGD003), Germany (location Gatersleben, DEU146), the Philippines (PHL129), Poland (POL003), Thailand (THA300), and Great Britain (GBR016)
Flash flood	Flood barriers, dikes, spurs, etc., floor plans for a quick water outflow, shelving above ground level [26,32,34]	- national genebank of India (IND001)
Tropical cyclone	Same as against storm surges and floods (e.g., embankment) [36,37], wind-resistant buildings [38], clearing of surroundings (e.g., cutting of trees in proximity to genebank) [39]	- international genebanks WorldVeg, Taiwan (TWN001) and CePaCT, Fiji (FIJ049)
Extratropical storm	Wind-resistant buildings, reinforcing/securing of roofs [40]	- international genebank SGSV (NOR051) - national genebanks of France (FRA139), Germany (DEU271) and Great Britain (GBR004, GBR016)
Tornado	Wind-resistant buildings, safe rooms (e.g., for seed storage rooms), reinforcing/securing of roofs [41,42]	- national genebanks of Canada (CAN025) and the United States of America (USA020, USA970, USA033)
Hailstorm	Hail-resistant roofs and windows [43]	- national genebanks of Colombia (COL017), Ethiopia (ETH085), and the United States of America (USA020)
Lightning	Lightning rod [44]	- national genebank of Pakistan (PAK001)
Wildfire	Fire breaks, non-combustible materials and fire-resistant structures, adequate vegetation [45,46]	- international genebank IITA, Nigeria (NGA039) - regional genebank SRGB, Zambia (ZMB030) - national genebanks of Kenya (KEN212) and Thailand (THA300)

^1^ The genebanks listed here are the most exposed locations per hazard amongst the sample, i.e., falling in the highest or second-highest class. This, however, does not mean that for other institutions, no infrastructural control measures are recommended. Source: adapted from Munich Re (2022) [19], risk control measures compiled from the literature cited in each risk measure cell.

**Table 4 plants-12-02874-t004:** Costs for in perpetuity conservation at selected CGIAR genebanks, per crop type, as a proxy for collection value.

Crop Type	CGIAR Genebank	Size of Collection		Present Values at Different Interest Rates				Collection Value	
		2001	2020	Reference Year ^1^		2021 ^2^		2021	
		[No. of Accessions]		2%	6%	2%	6%	2%	6%
				[US $ Per Accession]				[in Million US $]	
Common Bean	CIAT	31,400	32,347	47.1	12.9	76.8	21.0	2.484	0.680
Forages	CIAT	24,184	22,694	83.7	22.9	136.5	37.3	3.097	0.847
Wheat *	CIMMYT	154,912	146,505	22.7	6.3	42.4	11.8	6.208	1.734
Wheat **	CIMMYT	see above	see above	25.9	9.6	48.5	17.9	7.098	2.625
Maize *	CIMMYT	25,086	32,243	151.5	32.3	283.2	60.4	9.132	1.946
Maize **	CIMMYT	see above	see above	260.2	141.0	486.5	263.6	15.686	8.500
Sorghum	ICRISAT	36,721	42,352	47.4	14.3	81.1	24.5	3.434	1.038
Pearl Millet	ICRISAT	21,392	24,373	56.1	15.2	95.9	25.9	2.336	0.632
Chickpea	ICRISAT	17,250	20,764	47.8	14.4	81.8	24.6	1.699	0.510
Pigeonpea	ICRISAT	13,544	13,783	58.7	15.4	100.3	26.4	1.383	0. 363
Groundnut	ICRISAT	15,327	15,622	49.7	14.6	84.9	24.9	1.327	0.389
Rice, cultivated	IRRI	94,564	125,899	25.1	6.3	42.9	10.8	5.397	1.358
Rice, wild	IRRI	4568	5813	37.1	7.5	63.4	12.8	0.368	0.074

* without initial regeneration, ** with initial regeneration. ^1^ Reference year for CIAT 2000, CIMMYT 1996, ICRISAT and IRRI 1999. ^2^ Conversion to 2021 figures using OECD producer price index for OECD countries [63]. Source: adapted from Koo et al. (2003) [62]. Collection sizes for 2020 are from WIEWS (2020) [15].

**Table 5 plants-12-02874-t005:** Costs of conserving accessions in perpetuity at the IPK Gatersleben, per crop type.

Crop Type	Costs in Perpetuity (2018) [€ per Accession]
Wheat	13.00
Rye	13.00
Soybean—open air	11.98
Soybean—greenhouse	40.77
Chickpea—open air	12.26
Chickpea—greenhouse	29.57
Cabbage	41.88
Cauliflower	35.76
Lettuce	19.50

Source: own representation, adapted from Rabenau (2018) [64].

## Data Availability

The results of the natural hazard exposure analysis provided by Munich Re are made available in the annex and the supporting materials. Public datasets analysed stem from the WIEWS database (https://www.fao.org/wiews/data/ex-situ-sdg-251/overview/en/), the Worldbank’s database on the Worldwide Governance Indicator (https://databank.worldbank.org/source/worldwide-governance-indicators), and the database on the Fragile States Index by the Fund for Peace (https://fragilestatesindex.org/).

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
