# Peer review of "Genebanks at Risk: Hazard Assessment and Risk Management of National and International Genebanks"

_plants, 2023, doi:10.3390/plants12152874_

Round 1

Reviewer 1 Report

General Comments

This manuscript assesses 80 national and international genebanks located on all world regions against three hazards that could impact their operations: (1) natural hazards, (2) political risks, and (3) financial risks. Furthermore, it discusses risk management and mitigations strategies at genebanks regarding (1) infrastructure, (2) safety duplication, (3) insurances and (4) funds.

The paper provides some useful considerations about the global situation regrading plant genetic resources in genebanks. It has value for genebank managers and decisions makers. It is a paper that addresses corporate management issues and any other institution at the geographical genebank locations considered here will face the same challenges. The first part of paper is very general, and in my view the risks and approaches that are specific to genebanks are only addressed when it comes to discussing the risk management strategies. The biological and environmental factors come very much into play when you talk about active genebanks, but they receive little attention in this paper. The conclusion that the Svalbard Global Seed Vault is the most secure genebank operation is very questionable, because Svalabard is not an active genebank, it is simply an storage facility. The enormous genebank specific risks and uncertainties associated with regeneration, acquisition, characterization, data management, interaction with clients, receiving funding, training staff, adapting to changing technologies, and also to the global decreases in germplasm accessibility are not or only marginally considered in this paper.

The first part of the paper remains abstract. The specific environmental and political situations vary greatly among countries and within a country locations can be very different. What is presented is not surprising or ne. The grouping according to five continents seems questionable. It would be better if the authors would have followed the distinctions of world regions that is used in the context of the FAO Genetic Resources, because that would enhance the consistency regarding such general and global discussions. For example joining South and North America seems for political and also environmental reasons not very logical.

Since each genebank’s situation is so specific, a case by case consideration makes sense, and in my view a key message is that each genebank may take the general points addressed here and apply it to assess its own and specific risk status.

A central risk to genbanks is the staffing, and the knowledge transfer. The management issues when overseeing a genebank operation are not discussed.

Specific Comments

Line 9: Is this paper in fact providing a “scientific base”? In my view it strives to provide a rational base, but what constitutes a scientific base can be questioned an may depend on the understanding what science is. The Anglo-American concept of science differs from the European concept.

Lines 27-30: The authors refer here to the 2nd FAO Word Report from 2010. Later in the manuscript the authors use information form the 3rd SOW PGRFA Report which is in draft status. The global numbers are quite different, much lower, in the 3rd SOW PGRFA report. I suggest to use the recent numbers.

Line 77: Replace “measure” by “assessment” or “quantification”. I think this is what you have in mind.

Line 109: The numbers do never tell it all. 100 carrots and more difficult for a genebank than 1000 barleys. Perhaps comment on that.

Line 127: In my view it is not justified to list the SGSV in one sentence with active genebanks. The SGSV is nothing but a black-box storage facility. Even a commercial storage warehouse has more activity and is more vulnerable for that only.

Line 130: There are other biological sample storage facilities that may have even higher standards, e.g. for pathogenes.

Figure 1: I notice a large circle on the UK. Did you include the Arabidopsis collection? Does that make sense? Or is this the Kew Millennium Seed Bank? This is not an active genebank.

Line 169: Even if you do not discuss it, but the very relevant biological hazards affecting active genebanks need to be mentioned: pests, fungi, other diseases.

Table 1: The heading or footnote must at least briefly explain the ratings.

Lines 191-194: Active genebanks can not, or only partly, be compared to “ordinary commercial and industrial businesses”. Active genebanks needs to be assessed more like farming. I think this a major limitation of the assessment proposed here.

Figure 2: Please eliminate the background colour (same applies to Figure 1). It is very difficult to recognize the single dots.

Figure 3 and Figure 4: A grouping according to FAO SOW report regions would be better. It may be sufficient to show  one of the two figures. There is not much added value in showing both. Figure 5 caoukd also be integrated in this merged figure.

Line 249: If you single out NordGen, you must explain for what reason.

Line 250: You may mention her that there was an earthquake in 2008 just before the SGSV opened and a flooding of the hallways. (And then you could wonder about not enhancing the relationships with the neighbourhood with the Russians at Barentsburg. What if supplies by boat are cut off? That happened to the Russians last year.

Figure 5: This could be merged with Figure 3 and you could mark the dots for the international genebanks.

Section 3.2: For this paper it may be too much to present the difference between the two assessments of the WGI and the FSI. The correlation in Figure 8 shows that they are nearly identical. I suggest the authors merge both ratings inti one assessments and present that, or they chose one of the two.

Figures 6 and 7: One of the two is sufficient, or create a new map based on combination of the two assessments.

Line 332: The authors suggest that centralization results in strength. This could be questioned. It needs more discussion.

Line 357: The IT risks deserve a separate and very detailed discussion. The IT risks become increasingly relevant and are underestimated.

Table 2: Choosing a suitable location for a genebank in a given country is not mentioned as a control measure. For example the locations of Plant Gene Resources of Canada minimizes risk to pathogens and the dry air supports risk free seed handling, in addition the proximity to plant research facilitates the active utilization (see country report of Canada for 3rd SOW-PGRFA). The risks associated with staffing/trained personal is missing in the text

Table : Did you know that the planning of the IPK genebank building lay-out was impacted by Rudolf Mansfeld having experienced the bombing by the Allied Forces and burning of the herbarium in Berlin Dahlem in World War II, I think it was 1943? The IPK genebank building are stretched out and connected by long free standing corridors. If one segment burns, the others are not as exposed at the same time. A very intelligent planning based on horrible experience.

Table 3: The German IPK genebank experienced a disastrous flooding of all greenhouses and field in 1995.

Lines 385-388: This section needs  elaboration and point in my view at a key risk.

Line 409: Delete “Yes”.

Lines 417-418: This is not the case. The molecular assessments do not impact the back-up decisions. Plant Gene Resources of Canada uses exactly the same strategy as described for IPK.

Line 425 pp. You need to mention that a back-up or a duplication in another active genebank collection that goes beyond storage is also a very valid approach. This enhances knowledge about the accession duplicated as characterization and evaluations information added from various sites and the seeds are distributed and available. This is today of greatest relevance. We see that some genebanks are not accessible anymore, e.g. VIR, and we need the active duplication. Svalbard and other black-box back-ups are not of help in such situation. This needs discussion.

4550456: It is very true what you say about field genebanks.

Line 471: Add also the building of human capacity as a factor.

Section 4.2: If the Risk Transfer Strategy is not applicable, you may not need to mention it.

Section starting line 496:

This section is of financial nature. The correct value assessment of a genebank remains a mystery. Some accessions are simply irreplicable. This may need to be mentioned. The entire sections is very speculative and in my view is of very different nature that the previous sections. I suggest to shorten this section. Of cause, a financial insurance policy is fine. Bit this section goes to far into financial specificizes which are possibly not applicable across countries

Table 5: These exact numbers make me suspicious. I also would assume the costs for rye, being an out-crossing species, are much higher that the costs for wheat. I do not have much trust in these numbers.

Line 636: In my view you do not need t repeat the findings. The abstract does that. But you need to point at sharp and swift conclusions, at the “take home messages”.

Author Response

Dear Reviewer 1, 

thank you very much for your valuable suggestions and comments. We highly appreciated them and took them into consideration, whereever possible and feasible. Please find in the document attached our response to your review. 

With kind regards, 

Theresa Herbold and Johannes M.M. Engels

Reviewer 2 Report

This paper compares genebanks with respect to natural hazards, polititcal and finacial risks.The discussion on risk management strategies is informative, especially the discussion on insurance, which was interesting. . The paper needs substantial work on grammer.There are also a few areas that need further clarification or caveats. See specific comments below. 

Line 10- delete "here"

13- international is a more clear word then supranational. I would use it in all instances you have used supranational

Line 55-64: include under section 2.Objective and Methodology

Line 70: don't need journal year 

Line 100: add "of" after "question" 

line 120- replace "have been" with "were"

line 122: Sentence starting with "In total" should be deleted. 

Line 135-153- since you are describing what has already been done- you need to write in the past tense.

Line 358- do not know what you mean by "prospective"

Line 360: should be "duplication"

line 363: delete "allows to", add "s" to end of "mitigate"

Line 367 Table 2. Outage or malfunction- need to also have staff who are competent in making repairs

or assess to repair services

Table 3- Did you take into account the building codes of each country. The genebanks may already be built with risk control measures due to building codes.

Line 410-412- sentence does not seem relevant to paper.

Line 451: "ICARDA" not IRCADA (correct throughout paper)

Line 656- you need a caveat for the text above, where you define hazard risk. You need to acknowledge that some of these genebanks with high hazard risks, may have been constructed to withstand these hazards. For example, the USDA genebank in Colorado was constructed to withstand tornados and flooding. You also have mentioned SVSG is buried in a mountain.Genebanks need to identify risk based on hazards, but also existing risk management strategies such as construction, to come up with a final hazard risk assessment.This should be made clearer in the paper.

Conclusion- I would refer to your objectives as part of your conclusion.

Needs extensive revision

Author Response

Dear Reviewer 2, 

thank you for your valuable comments and suggestions. We highly appreciated them and incorporated nearly all of them. Please be informed that the manuscript was reviewed now by a native speaker, so that all language and grammar issues should be solved. In the document attached you can find our detailed response to your review. 

With kind regards, 

Theresa Herbold and Johannes M.M. Engels
